# Deglycosylated Azithromycin Attenuates Bleomycin-Induced Pulmonary Fibrosis via the TGF-β1 Signaling Pathway

**DOI:** 10.3390/molecules26092820

**Published:** 2021-05-10

**Authors:** Hao Ruan, Shaoyan Gao, Shuangling Li, Jiaoyan Luan, Qiuyan Jiang, Xiaohe Li, Huijun Yin, Honggang Zhou, Cheng Yang

**Affiliations:** 1State Key Laboratory of Medicinal Chemical Biology, College of Pharmacy and Key Laboratory of Molecular Drug Research, Nankai University, Tianjin 300000, China; ruanhao8958@mail.nankai.edu.cn (H.R.); ShaoyanGao339@163.com (S.G.); 2120191206@mail.nankai.edu.cn (S.L.); 13553118700@163.com (J.L.); Tiancha18822188139@163.com (Q.J.); lixiaohe908@163.com (X.L.); 2High-throughput Molecular Drug Screening Centre, Tianjin International Joint Academy of Biomedicine, Tianjin 300070, China; 3China Resources Pharmaceutical Group Limited, Beijing 100000, China

**Keywords:** deglycosylated azithromycin, pulmonary fibrosis, transforming growth factor-β1, epithelial to mesenchymal transition, inflammation

## Abstract

Idiopathic pulmonary fibrosis (IPF) is a progressive, life-threatening lung disease characterized by the proliferation of myofibroblasts and deposition of extracellular matrix that results in irreversible distortion of the lung structure and the formation of focal fibrosis. The molecular mechanism of IPF is not fully understood, and there is no satisfactory treatment. However, most studies suggest that abnormal activation of transforming growth factor-β1 (TGF-β1) can promote fibroblast activation and epithelial to mesenchymal transition (EMT) to induce pulmonary fibrosis. Deglycosylated azithromycin (Deg-AZM) is a compound we previously obtained by removing glycosyls from azithromycin; it was demonstrated to exert little or no antibacterial effects. Here, we discovered a new function of Deg-AZM in pulmonary fibrosis. *In vivo* experiments showed that Deg-AZM could significantly reduce bleomycin-induced pulmonary fibrosis and restore respiratory function. Further study revealed the anti-inflammatory and antioxidant effects of Deg-AZM *in vivo*. *In vitro* experiments showed that Deg-AZM inhibited TGF-β1 signaling, weakened the activation and differentiation of lung fibroblasts, and inhibited TGF-β1-induced EMT in alveolar epithelial cells. In conclusion, our findings show that Deg-AZM exerts antifibrotic effects by inhibiting TGF-β1-induced myofibroblast activation and EMT.

## 1. Introduction

Idiopathic pulmonary fibrosis (IPF) is a chronic, progressive, and fibrous interstitial lung disease with unknown etiology [1]. A common feature of pulmonary fibrosis is excessive proliferation of fibroblasts and the accumulation of extracellular matrix (ECM), which subsequently causes alveolar structure destruction and fibrotic foci formation. IPF is a severe disease with a median survival time of only 2.5 years from diagnosis [2]. Currently, only two drugs, nintedanib and pirfenidone, are approved specifically for IPF treatment, but their efficacies are limited, and they cannot prolong patient survival time [3]. Therefore, it is particularly important to explore new potential therapeutic drugs.

Although much of the pathogenesis of IPF remains to be elucidated, the major paradigm is that aberrant injury-associated remodeling of epithelial cells induces myofibroblast activation through multiple cytokines and inflammatory factors, and these myofibroblasts in turn produce excessive ECM and promote fibrogenesis [4,5,6,7]. Among these cytokines, transforming growth factor-β1 (TGF-β1) is the strongest profibrotic mediator because it simultaneously contributes to epithelial cell injury and myofibroblast differentiation. Knockdown of the TGF-β type II receptor attenuates bleomycin-induced pulmonary fibrosis [8], and an antibody targeting TGF-β1 protects against the profibrotic phenotype *in vivo* [9]. Moreover, TGF-β1 is considered to directly drive epithelial to mesenchymal transition (EMT) and to perpetuate this event in the lung [10]. It has been proposed that epithelial cells undergo EMT to give rise to fibroblasts, which substantially contributes to the number of myofibroblasts [11,12]. Although the role of EMT in pulmonary fibrosis has been controversial in recent years [13], high expression levels of stromal markers in epithelial cells indeed enhance fibrosis progression [14]. Consequently, inhibiting TGF-β1-induced fibroblast activation and EMT may be an effective strategy for IPF treatment.

Azithromycin is a member of the macrolide family and exerts both immunomodulatory and antibacterial effects. A study has shown that azithromycin alleviates bleomycin-induced pulmonary fibrosis by downregulating the inflammatory response and oxidative bursts [15]. A retrospective study also demonstrated that azithromycin was a safe and well-tolerated treatment in individuals with IPF [16]. However, it is still not clear whether azithromycin confers this benefit by antibacterial or other effects. Previously, we obtained deglycosylated azithromycin (Deg-AZM, Figure 1A) by removing both glycosyls and demonstrated that this compound has little or no antibacterial effects [17]. Here, we show that Deg-AZM exerted antifibrotic effects independent of antibacterial effects. Deg-AZM attenuated bleomycin-induced pulmonary fibrosis, the inflammatory response and oxidative stress *in vivo*. Further studies showed that Deg-AZM exerted antifibrotic effects by inhibiting myofibroblast activation and EMT induced by TGF-β1. This study provides a potential basis for the use and clinical development of Deg-AZM for IPF treatment.

## 2. Results

### 2.1. Deglycosylated Azithromycin Attenuates Bleomycin-Induced Pulmonary Fibrosis in Mice

To determine the antifibrotic effects of Deg-AZM *in vivo*, we established a pulmonary fibrosis model by intratracheally administering bleomycin to C57BL/6 mice (Figure 1B). Different doses of Deg-AZM (50, 100, 200 mg/kg) were applied beginning on the 7th day after bleomycin injury, while nintedanib (100 mg/kg) and pirfenidone (200 mg/kg) were used as positive controls. We then evaluated the effect of Deg-AZM on the recovery of respiratory function. As shown in Figure 1, there were reductions in forced vital capacity (FVC, Figure 1C) and dynamic compliance (Cdyn, Figure 1D) and increases in inspiratory resistance (Ri, Figure 1E) and expiratory resistance (Re, Figure 1F) in the model group. Deg-AZM treatment markedly restored respiratory functions. In addition, H&E staining was performed to evaluate the degree of pulmonary fibrosis. The model group displayed severe fibrotic foci formation, while Deg-AZM treatment reduced fibrosis, and the high dose group (200 mg/kg) exhibited similar results to nintedanib and pirfenidone (Figure 2A). This result was further verified in the quantified lung sections results (~54% decreased, Figure 2B). Furthermore, Masson’s trichrome staining (Figure 2C) and hydroxyproline (HYP) analysis revealed serious collagen deposition in the model group, and Deg-AZM treatment reduced the collagen content, especially for the high dose group (~43% decreased compared to BLM group, Figure 2D). These results demonstrate that Deg-AZM can attenuate bleomycin-induced pulmonary fibrosis.

### 2.2. Deglycosylated Azithromycin Suppresses the Inflammatory Response and Oxidative Stress in Vivo

Inflammation occurs in the early stages of IPF [18]. Therefore, we used BLM intratracheal instillation to induce an inflammatory response in mice and administered different doses of Deg-AZM (50, 100, 200 mg/kg) on the day after BLM administration. Nintedanib (100 mg/kg) and pirfenidone (200 mg/kg) were used as positive controls. H&E staining of pathological sections of the left lung and H&E staining of inflammatory cells in bronchoalveolar lavage fluid (BALF) showed that Deg-AZM reduced inflammatory cell infiltration, and the therapeutic effect of Deg-AZM was better than that of nintedanib and pirfenidone (Figure 3A,B). We then counted the inflammatory cells in BALF and found that Deg-AZM high dose significantly reduced the numbers of neutrophils (~59% decreased, Figure 3C), lymphocytes (~45% decreased, Figure 3D) and macrophages (~67% decreased, Figure 3E). We also measured the concentration of inflammatory cytokines in BALF, including interleukin (IL)-4 and IL-1β, using ELISA. The levels of these cytokines were elevated in response to bleomycin administration in the model group but were reduced by Deg-AZM treatment (Figure 3F, G). Finally, we tested the antioxidant capacity of Deg-AZM. As shown in Figure 3H,I, bleomycin administration decreased the levels of superoxide dismutase (SOD) and the total antioxidant capacity (T-AOC), and Deg-AZM treatment improved SOD levels and the T-AOC. Therefore, these data indicate that Deg-AZM can inhibit the bleomycin-induced inflammatory response and enhance antioxidant capacity.

### 2.3. Deglycosylated Azithromycin Inhibits TGF-β1-Induced Fibroblast Proliferation and Migration

In response to injury, the proliferation and migration of activated fibroblasts occurs in the initial stage of tissue repair, which contributes to the development of lung fibrosis [19,20]. To determine whether Deg-AZM affects TGF-β1-induced fibroblast proliferation and migration, NIH-3T3 cells were cultured with or without 5 ng/mL TGF-β1 and different doses of Deg-AZM for 24 h. The proliferation of NIH-3T3 cells was measured by MTT assays. We found that Deg-AZM showed no obvious toxicity to normal cells, with an IC_50_ higher than 1 mM (Figure 4A,B). We also used a wound healing assay to verify the effect of Deg-AZM on TGF-β1-induced fibroblast migration, and the results showed that Deg-AZM could inhibit fibroblast migration (Figure 4C).

### 2.4. Deglycosylated Azithromycin Attenuates TGF-β1-Induced Myofibroblast Activation

To further investigate the effect of Deg-AZM on TGF-β1-induced myofibroblast activation, we treated NIH-3T3 cells with TGF-β1 (5 ng/mL) and different doses of Deg-AZM to evaluate the gene expression levels of fibroblast activation markers. As shown in Figure 5A–G, the expression levels of the profibrotic factors, fibronectin (Fn), collagen I (Col1) and α-smooth muscle actin (α-SMA) were markedly increased by TGF-β1 stimulation, and Deg-AZM decreased the mRNA (Figure 5A–C) and protein levels (Figure 5D–G). The immunofluorescence assay results support a similar conclusion (Figure 5H). These results indicate that Deg-AZM can suppress TGF-β1-induced myofibroblast differentiation and collagen expression *in vivo*.

TGF-β1 signaling is a master switch in fibrogenesis, and its downstream transcription factor Smad3 plays a critical role in signal activation. Therefore, we verified whether Deg-AZM could inhibit Smad3 activation. Moreover, TGF-β1 initiates the mitogen-activated protein kinase (MAPK) family by activating downstream Erk1/2, p38, and Akt [21]. Thus, we also examined the effect of Deg-AZM on TGF-β1/MAPK signaling. The results showed that Deg-AZM inhibited the TGF-β1-induced phosphorylation of Akt, Smad3, p38 and Erk1/2 (Figure 5I–M). These data suggest that Deg-AZM can block TGF-β1-induced Smad3 and MAPK signaling and further suppress myofibroblast activation.

### 2.5. Deglycosylated Azithromycin Inhibits TGF-β1-Induced EMT in Epithelial Cells

EMT is the process by which an epithelial cell transforms into a mesenchymal cell phenotype. This process is closely associated with the development of pulmonary fibrosis, although it has been controversial in recent years [22]. To investigate whether Deg-AZM can inhibit the EMT process, we induced EMT by treating A549 cells with TGF-β1. The results showed that Deg-AZM significantly increased the epithelial marker E-cadherin (Deg-AZM 40 μM+T vs. TGF-β1, *p* = 0.0025, Figure 6A) and decreased the mesenchymal markers N-cadherin (Deg-AZM 40 μM+T vs. TGF-β1, *p* = 0.0001, Figure 6B) and Vimentin (Deg-AZM 40 μM+T vs. TGF-β1, *p* = 0.0039, Figure 6C). This result was further confirmed by Western blotting and immunofluorescence assays (Figure 6D–I). Therefore, Deg-AZM can inhibit the TGF-β1-induced EMT process in alveolar epithelial cells.

### 2.6. Deglycosylated Azithromycin Inhibits the Fibrogenic Activation of Pulmonary Fibroblasts in Vivo

To further verify whether Deg-AZM can attenuate the activation of lung fibroblasts *in vivo*, we extracted protein and RNA from the lung tissues of C57BL/6 mice and analyzed the expression of Fn, Col1 and α-SMA by Western blotting and real-time PCR. As expected, the protein levels of Fn, Col1 and α-SMA in the lung tissues of mice treated with Deg-AZM were lower than those in the bleomycin model group (Figure 7A–D). Similarly, the RNA expression levels of Fn (*p* = 0.0004), Col1 (*p* = 0.0047) and α-SMA (*p* = 0.0005) in the Deg-AZM high dose group were significantly lower than those in the bleomycin model group. We also performed immunohistochemical analysis of mouse lung slices. The results confirmed the previous observations (Figure 7H–J). Furthermore, we detected the TGF-β1/Smad and non-Smad signaling activation. As shown in Figure 7 K–O, Deg-AZM decreased the phosphorylation level of Akt, Smad3, p38 and Erk1/2 in lung tissues. In conclusion, Deg-AZM downregulates the expression of Fn, Col1 and α-SMA in lung tissues, thereby reducing extracellular matrix deposition and bleomycin-induced lung fibrosis in mice.

### 2.7. Deglycosylated Azithromycin Inhibits the EMT Phenotype in Vivo

We further evaluated the effect of Deg-AZM on the progression of EMT in a BLM-induced C57BL/6 mouse lung fibrosis model. Similar to our *in vitro* results, the Western blot results showed that Deg-AZM increased the expression of E-cadherin in tissues and decreased the expression of N-cadherin and Vimentin (Figure 8A–D). In addition, RT-PCR analysis of lung tissues showed that Deg-AZM upregulated E-cadherin (Deg-AZM 200 mg/kg vs. BLM, *p* = 0.0079, Figure 8E) and downregulated N-cadherin (Deg-AZM 200 mg/kg vs. BLM, *p* = 0.0006, Figure 8F) and Vimentin (Deg-AZM 200 mg/kg vs. BLM, *p* = 0.0134, Figure 8G). Similarly, immunohistochemical analysis showed that the number of E-cadherin-positive cells increased, and the number of N-cadherin and Vimentin-positive cells decreased (Figure 8H–J). Since BLM-induced pulmonary fibrosis mice exhibit high levels of TGF-β1 [23], and TGF-β1 will in turn enhance the EMT process, we also measured the TGF-β1 protein level in lung tissues. The result showed that the TGF-β1 level was indeed significantly elevated after BLM stimulation, while Deg-AZM slightly decreased the TGF-β1 level (Appendix A). These findings suggest that Deg-AZM inhibits EMT *in vivo*.

## 3. Discussion

IPF is a chronic progressive interstitial lung disease of unknown origin that can quickly lead to death [24]. IPF is characterized by abnormal and uncontrolled fibroblast proliferation, distortion of the lung structure caused by chronic and repetitive damage to alveolar epithelial cells, and subsequent dysregulated repair processes [25]. Injured type II alveolar epithelial cells repeatedly release a variety of inflammatory factors and cytokines, which promote the excessive activation of fibroblasts, leading to an imbalance in cell regeneration and scar repair and ultimately IPF [26]. Therefore, inhibiting fibroblast activation is critical for IPF treatment. Azithromycin is a macrolide with immunomodulatory properties and anti-inflammatory effects that is widely used in respiratory infection treatment. It has been suggested that azithromycin is beneficial in treating IPF and acute exacerbation of IPF (AE-IPF) [16,27], and researchers believe this benefit is associated with antibiotic activity. Here, we examined the antifibrotic effect of Deg-AZM, which we previously demonstrated to have little or no antibiotic activity compared with that of azithromycin. Both *in vivo* and *in vitro* results verified that Deg-AZM could attenuate fibrogenesis by suppressing TGF-β1-induced fibroblast activation and EMT (Figure 9).

At present, the BLM-induced pulmonary fibrosis model is the most classic model for exploring the pathogenesis of IPF and verifying the effectiveness of antifibrotic drugs [28]. In this study, we established a BLM-induced pulmonary fibrosis and inflammation model to evaluate the antifibrotic effects of Deg-AZM. Deg-AZM treatment significantly restored pulmonary function, improved the fibrosis score and decreased the hydroxyproline content in a dose-dependent manner, and some of these effects were better than those of two approved drugs, nintedanib and pirfenidone. In the early stage of IPF (acute lung injury), Deg-AZM reduced the number of inflammatory cells in BALF and decreased the secretion of inflammatory cytokines, including IL-4, IL-1β and IFN-γ, which have been demonstrated to be increased in IPF patients [29,30]. Deg-AZM also showed promising antioxidant capacity by improving SOD levels and the T-AOC of lung tissues. Previous study has investigated the anti-inflammation and fibrosis effect of AZM in BLM-induced pulmonary fibrosis model [15,31]. Researchers gave mice with AZM (3.5 mg/kg) every other day after BLM injection for analysis, and found alleviated fibrosis formation, enhanced antioxidative properties and decreased profibrotic cytokines (including IL-1β) secretion. These findings are in line with us. Since we remove the antibiotic effect of AZM, the application dose can be raised without showing obvious toxicity, and showed better effects than AZM. Together, these results indicate that Deg-AZM protects against BLM-induced inflammation and fibrosis, and laid the foundation for us to further study its pharmacological mechanism.

Although the pathogenesis of pulmonary fibrosis remains unclear, TGF-β1 is currently considered to be one of the most effective profibrotic cytokines [32]. In IPF patients, TGF-β1 is mainly secreted by lung macrophages and alveolar epithelial cells and can stimulate the proliferation and activation of fibroblasts, promote myofibroblast formation and ECM deposition, and induce EMT in alveolar epithelial cells [33]. Thus, we used a TGF-β1-induced NIH-3T3 cell model for our subsequent experiment. Deg-AZM inhibited the TGF-β1-induced proliferation and migration of NIH-3T3 cells without showing obvious cytotoxicity. Further study showed that Deg-AZM downregulated TGF-β1-induced Smad3, Erk, P38 and Akt phosphorylation and blocked the activation of these pathways. As expected, the *in vivo* data revealed that Deg-AZM inhibited ECM deposition and TGF-β1 signaling activation. These results indicate that Deg-AZM treatment can be effective in the local microenvironment of lung tissue and can maintain the balance in the degradation of ECM components, which is of great importance for IPF treatment.

In addition to the proliferation and differentiation of resident lung fibroblasts, myofibroblasts may also be derived partially from fibroblast EMT. In lung tissues, approximately 33% of fibroblasts originate from cells undergoing EMT [34]. Therefore, we also studied the effect of Deg-AZM on EMT inhibition. We used TGF-β1 to induce EMT-like changes in A549 cells and measured the expression of EMT markers such as E-cadherin, N-cadherin and Vimentin [35]. The results showed that Deg-AZM treatment downregulated Vimentin and upregulated the expression of E-cadherin and further inhibited TGF-β1-induced EMT. An *in vivo* experiment also showed that Deg-AZM repressed the EMT phenotype, which indicated that the antifibrotic effect of Deg-AZM is at least partly due to the inhibition of EMT. Moreover, we analyzed the protein level of TGF-β1 in lung tissues. BLM-induced mice fibrosis model exhibit excessive secretion of multiple profibrotic cytokines, especially for TGF-β1. Inhibiting TGF-β1 hyperaction showed prolonged effect on BLM-induced mice fibrosis [36]. We found that a high dose of Deg-AZM could slightly decrease the expression of TGF-β1. This will help in alleviating EMT process induced by TGF-β1 overexpression. These results suggest that Deg-AZM, the parent structure of azithromycin, exhibits antifibrotic effects by suppressing TGF-β1-induced EMT.

The recently emerged severe acute respiratory syndrome coronavirus 2 (SARS-CoV-2) can result in an excessive host inflammatory response characterized by the hyperproduction of cytokines. The clinical symptoms of SARS-CoV-2 become more severe with age and some complications, such as hypertension and diabetes [37], are also common in IPF, suggesting that fibrosis may be a substantial risk in some patients after SARS-CoV-2 infection. Our findings suggest that Deg-AZM may have the potential to suppress the overexpression of cytokines and inhibit fibrosis after SARS-CoV-2 infection.

## 4. Materials and Methods

### 4.1. Antibodies and Reagents

Deg-AZM (≥98.0%) was manufactured by chemical laboratory, College of Pharmacy, Nankai University. Recombinant Human TGF-β1 was purchased from PEPROTECH (USA). Antibodies to α-SMA, Collagen I, Fibronectin, P-Smad3/Smad3, P-Akt/Akt, P-ERK/ERK, P-P38/P38 were from Affinity Biosciences (Cincinnati, OH, USA). GAPDH, E-cadherin, Vimentin, N-cadherin antibodies were from Cell Signaling Technology (Danvers, MA, USA). Goat pAb to Rb IgG (HRP) and Rb pAb to Ms IgG were from ImmunoWay (Shanghai, China). TRIzol reagent was acquired from Ambion Life Technology (Shanghai, China). DEPC Treated H_2_O and Rnase Away H_2_O were purchased from Life Technologies. UNICON^®^ Qpcr SYBR Green Master Mix was from YEVSEN (Shanghai, China). BLM was from Nippon Kayaku (Tokyo, Japan). Hematoxylin-eosin solution and Masson’s Trichrome Stain Kit were from Solarbio.

### 4.2. Cell Culture

Mouse fibroblast cells (NIH3T3, purchased from ATCC) were grown in DMEM (KeyGEN BioTECH, Nan Jing, China), supplemented with 10% fetal bovine serum (FBS, ExCell Bio, Shanghai, China). A549 cells were cultured in RPMI medium 1640 (contain PS, Solarbio, Beijing, China), both supplemented with 10% FBS (Biological Industries, Kibbutz Beit-Haemek, Israel) at 37 °C with 5% CO_2_. After 24 h of serum starvation, 5 ng/mL TGF-β1 was supplemented to induce lung fibroblast activation and EMT. In most experiments, cells were maintained in media supplemented with TGF-β1 with or without Deg-AZM for 24 h to evaluate the effect of Deg-AZM on cell proliferation and EMT. When studying the effects of Deg-AZM on TGF-β1/Smad3, TGF-β1/MAPK (Erk1/2 and P38), PI3K/Akt and other signaling pathways, the cells were preincubated for 24 h with Deg-AZM before treatment with exogenous TGF-β1 induce for 30 min.

### 4.3. Animals

C57BL/6 mice at 7–8 weeks were obtained from Vital River Laboratory Animal Technology Co., Ltd. (Beijing, China). Mice were housed in a room at a temperature of 22 to 26 °C and a humidity between 40% and 70%, with 12 h light/dark cycle. Mice were free to eat and drink. All animal care and experimental procedures complied with guidelines approved by the Institutional Animal Care and Use Committee (IACUC) of Nankai University (Permit No. SYXK 2014-0003). Animal studies are reported in compliance with the ARRIVE guidelines (Kliment and Oury, 2010; McGrath and Lilley, 2015).

### 4.4. Bleomycin Administration

The intratracheal BLM mice pulmonary fibrosis model was performed as described previously [38]. First, the mice were anesthetized by intraperitoneal injection and weighed. Then the mouse belt was fixed to the operating table, the neck of the mouse was disinfected with 75% alcohol and a wound of about 1 cm was cut with a scalpel, and the muscle tissue was carefully stripped to expose its trachea clearly. At a dose of 2 U/kg, we used a sterile insulin syringe (with about 10 units of air column) to suck the corresponding volume of the prepared BLM solution, and slowly and parallelly inject the BLM into the mouse trachea from the cartilage ring of the mouse trachea. We put the mouse body in an upright position immediately and beat the back of the mouse several times (in order to make the BLM evenly distributed in the lung tissue). The control group was injected with physiological saline in the same way and at the same dose. Mice in each group were administered on the 7th day of modeling, and anatomical samples were taken on the 15th day to evaluate the degree of pulmonary fibrosis and analyze the therapeutic effects of the drugs. In the sham operation group, the same amount of saline was injected intratracheally by the same method. For fibrosis model, 70 mice were divided into seven groups, with 10 mice per group randomly assigned: blank control group, BLM model group, BLM+ nintedanib positive drug group (100 mg/kg), BLM+ pirfenidone positive drug group (200 mg/kg), BLM+ low-dose Deg-AZM group (50 mg/kg), BLM+ medium-dose Deg-AZM group (100 mg/kg), and BLM+ high-dose Deg-AZM group (200 mg/kg); no uninjured control group was included. The mice in the latter group were treated with 50 mg/kg, 100 mg/kg, 200 mg/kg, which was suspended in a 5% DMSO, 95% saline solution, while mice in the control group and BLM group were received an equal volume of vehicle (5% DMSO, 95% saline). Deg-AZM was intragastrically administered daily for 1 week starting 7 days after BLM application. The control and model groups received an equal volume of vehicle using the same schedule and route of administration. After BLM treatment on the 14th day, mice were sacrificed for subsequent experiments including hydroxyproline content determination etc.

For the inflammation model, 70 mice were divided into seven groups, with 10 mice per group assigned randomly: blank control group, BLM model group, BLM+ nintedanib positive drug group (100 mg/kg), BLM+ pirfenidone positive drug group (200 mg/kg), BLM+ low-dose Deg-AZM group (50 mg/kg), BLM+ medium-dose Deg-AZM group (100 mg/kg), and BLM+ high-dose Deg-AZM group (200 mg/kg); no uninjured control group was included. Deg-AZM was intragastrically administered daily for 1 week starting 1 day after the BLM injury. Mice were sacrificed on the 8th day after bleomycin administration for the evaluation of inflammation.

### 4.5. Pulmonary Function Testing

Mice were anesthetized and tracheotomized below the larynx, and intubated with a Y-type tracheal cannula. After the surgery, the mice were placed in a supine position with their whole body inside the plethysmographic chamber to analyze pulmonary function using the Anires2005 system (Beijing Biolab, Beijing, China). One branch of the Y-type tracheal cannula was connected to a venthole on the chamber wall, and the venthole was connected to a ventilator from the outside of the chamber by polyethylene tubing. Natural air was given to the animal through the ventilator, with a rate of 80 breaths/min. The other branch of the cannula was connected to the pressure-detecting channel of the Anires 2005 system, and the peak inspiratory pressure was maintained at 10–16 cm H_2_O. By detecting the ventilation-associated pressure change inside the chamber, the Anires2005 system automatically calculates and displays pulmonary function parameters, such as forced vital capacity (FVC), dynamic compliance (Cdyn), inspiratory resistance (Ri), and expiratory resistance (Re).

### 4.6. Hydroxyproline Measurement

The right lung tissues of the mice in the bleomycin-induced pulmonary fibrosis model were isolated for determining the Hyp levels according to a modified method described by Dong Y et al. [39]. The right lungs of mice were isolated, placed in 5 mL ampoules, dried, acid hydrolyzed, adjusted to pH 6.5–8.0, filtered, and adjusted to a total volume of 10 mL with 1× PBS. The sample was treated according to the kit used to measure hydroxyproline content. Then, the samples were cooled, 200 μL of each sample was transferred to 96-well plates in triplicate, and the absorbance was measured at 577 nm.

### 4.7. Histological Examination

The left lung was fixed in 10% formalin for 24 h, dehydrated and embedded in paraffin. Tissue sections with thickness of 4 μm were incubated for 4 h at 60 °C, which were stained with hematoxylin and eosin (H&E) and Masson’s trichrome. Quantification of pulmonary fibrosis was performed as described previously [40]. Images were tested by Image-Pro Plus Version 6.0, which can demarcate the entire lung area and automatically calculate total pixel Pw of the region and then calculate total pixel Pf of the fibrosis region (fibrosis ratio = fibrosis area total pixel Pf/total lung total pixel Pw).

### 4.8. Immunohistochemistry Staining

For immunohistochemistry assays, paraffin-embedded lung tissue sections were deparaffinized and antigen-retrieved. The following experimental steps were performed according to the instructions of the UltraSensitiveTM s-p immunohistochemical staining kit (MAIXIN.BIO, Fuzhou, China). Sections were stained with DAB solution (MAIXIN.BIO) and counterstained using haematoxylin. Eight images per slide and six slides per treatment group were quantified by Image Pro Plus 6.0 for integral optical density (IOD) and the positive area (Area). The mean optical density was calculated to evaluate the expression of α-SMA, collagen I, fibronectin.

### 4.9. ELISA for the Detection of Inflammatory Factors

The levels of IL-4, IL-1βand IFN-γ in bronchoalveolar lavage fluid (BALF) of bleomycin-induced inflammation model groups were determined by the enzyme-linked immunosorbent assay (ELISA) kit. Strictly follow the instructions of the ELISA kit (Jianglaibio, Shanghai, China).

### 4.10. Measurement of Oxidative Stress

The lung homogenates were centrifuged at 4 °C, and the supernatant was retained for testing. The levels of superoxide dismutase (SOD) and total antioxidant capacity (T-AOC), in lung tissue were measured according to the instructions of the detection kits (Solarbio, Beijing, China).

### 4.11. Bronchoalveolar Lavage

The mice were anesthetized and the left and right lungs are separated, and lavage was performed through a cannula connected to a syringe, which works as a tracheal cannula in the airway. BALF was collected by washing the lung through a tracheal intubation. The lungs were washed three times, and each time 1 mL 1× PBS was used. Total fluid recovery was about 90% per time. BALF was centrifuged at 3000 rpm for 10 min. The lysate was centrifuged at 3000× rpm for 10 min at room temperature, and the supernatant was removed into a 2 mL EP tube. The cell precipitation was suspended again with 200 μL 1× PBS. A 50 μL cell suspension was used for the smear, followed by HE staining, and the count of neutrophils, macrophages, and lymphocytes was performed under an optical microscope. The total cell count was performed by the Countstar automated cells Counting tool.

### 4.12. Quantitative Real-Time PCR (qRT-PCR)

Primers were used to determine the mRNA expression level of genes by RT-PCR according to the previously described protocol [41]. RNA was isolated from NIH-3T3 cells and tissue using TRIzol Reagent. The cDNA was synthesized with FastKing gDNA Dispelling RT SuperMix and prepared for RT-PCR. The qRT-PCR was carried by UNICON® Qpcr SYBR Green Master Mix according to the manufacturer’s instruction. The primers optimized for real-time PCR assays are listed in Table 1.

### 4.13. Western Blotting Analysis

The proteins were extracted from lung tissues or cells following standard protocols, as described previously [42]. Protein was extracted from lung tissue homogenates or cells using Radio-Immunoprecipitation Assay (RIPA) lysis buffer containing phenylmethylsulfonyl fluoride (PMSF) and sodium fluoride (NaF). After electrophoresis and membrane transfer, the following primary antibodies were used to explore the Western blot: GAPDH, α-SMA, Collagen I, Fibronectin, E-cadherin, N-cadherin, Vimentin, Phospho-Smad3, Smad3. Phospho-Erk 1/2 (Thr202/Tyr204), Erk 1/2, Phospho-pan-Akt 1/2/3 (Ser473), pan-Akt 1/2/3, Phospho-p38 MAPK (Thr180/Tyr182), and p38 MAPK. The secondary antibodies were goat anti-rabbit or goat anti-mouse horseradish peroxidase-conjugated antibodies. The protein bands were visualized using an enhanced chemiluminescence system (Affinity Biosciences, USA). GAPDH was used as the loading control. All original blots are uploaded and can be found in Appendix A.

### 4.14. Immunofluorescence Staining

NIH3T3 and A549 cells were fixed in 4% paraformaldehyde and stained with specific primary antibodies, including those against α-SMA, N-cadherin and Vimentin, at 4 °C overnight. They were then incubated with FITC-conjugated anti-rabbit IgG (Solarbio, Beijing, China). The nuclei were stained with DAPI. Representative micrographs were observed using a confocal laser scanning microscope (Leica, Wetzlar, Germany).

### 4.15. Cell Viability Analysis

NIH3T3 cells were initially seeded in 96-well plates prior to incubation for 24 h at 37 °C. The cell culture media was then replaced by complete media containing the indicated concentrations (0 to 800 μM) of Deg-AZM prior to incubation for specific times (24 or 48 h). After incubation as indicated, 15 μL of Thiazolyl Blue Tetrazolium Bromide (MTT, 5 mg/mL, KeyGENE BioTech, Nanjing, China) was administered to cells, followed by incubation for 4 h at 37 °C following the manufacturer’s instructions. Then, 200 μL DMSO was added per well, and the absorbance value was determined at 570 nm.

### 4.16. Wound-Healing Assay

NIH-3T3 cells were seeded in a six-well plate and treated with Deg-AZM in the presence or absence of TGF-β1 (5 ng/mL) for the indicated period. Three parallel lines were drawn on the underside of each well to demarcate the wound areas for analysis. Before inflicting the wound, the cells were fully confluent. A scratch was made in the center of the culture well using a sterile 200 µL micropipette tip. The wounds were observed using an inverted optical microscope, and multiple images were obtained at areas flanking the intersections of the wound and the marker lines after the scratch at 0, 12, 24 and 48 h. Images were analyzed using ipwin 32 software.

### 4.17. Data and Statistical Analysis

All statistical analyses were performed using Graphpad prism 7.0 software as the means ± SD (GraphPad Software, Inc., La Jolla, CA, USA). The comparisons were done by one-way analysis of variance (ANOVA) followed by the Tukey–Kramer test to identify significant differences between groups. *p* < 0.05 was considered to be statistically significant.

## 5. Conclusions

In conclusion, our results show that Deg-AZM treatment can protect against bleomycin-induced lung fibrosis and acute lung injury in mice. Deg-AZM can suppress TGF-β1-induced fibroblast activation and EMT. These findings suggest that Deg-AZM may be a candidate for IPF treatment and can provide additional potential options for IPF patients.

## Figures and Tables

**Figure 1 molecules-26-02820-f001:**
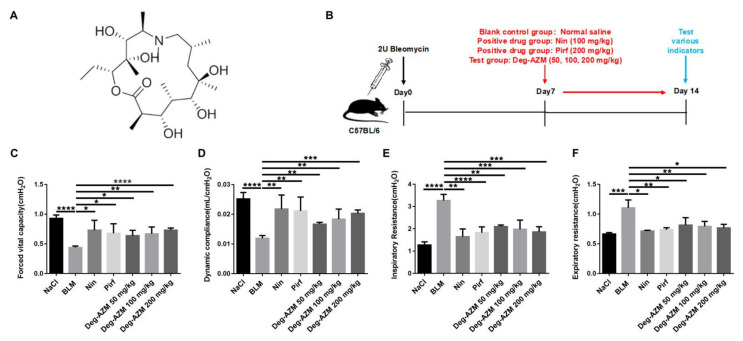
Deglycosylated azithromycin attenuates bleomycin-induced pulmonary fibrosis in mice. (**A**) Chemical structure of deglycosylated azithromycin. (**B**) Dosing regimen in BLM-induced pulmonary fibrosis model. (**C**) Forced vital capacity (FVC). (**D**) Dynamic compliance (Cdyn). (**E**) Inspiratory resistance (Ri). (**F**) Expiratory resistance (Re) was compared among different groups. Data are presented as the means ± SD (*n* = 8). Comparisons between groups were achieved using one-way analysis of variance (ANOVA). * *p* < 0.05, ** *p* < 0.01, *** *p* < 0.001, **** *p* < 0.0001.

**Figure 2 molecules-26-02820-f002:**
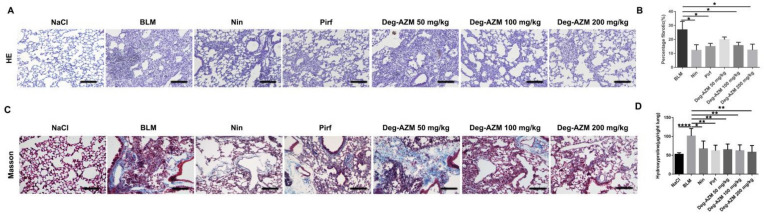
Deglycosylated azithromycin attenuates bleomycin-induced pulmonary fibrosis in mice. (**A**) Lung tissue sections were stained with hematoxylin-eosin (HE). (**B**) Statistics of lung fibrosis area among groups. (**C**) Lung tissue sections were stained with Masson Trichrome staining. (**D**) Hydroxyproline (HYP) contents of lung tissues in mice. Scale bar = 50 μm. Data are presented as the means ± SD (*n* = 8). Comparisons between groups were achieved using one-way analysis of variance (ANOVA). * *p* < 0.05, ** *p* < 0.01, *** *p* < 0.001, **** *p* < 0.0001.

**Figure 3 molecules-26-02820-f003:**
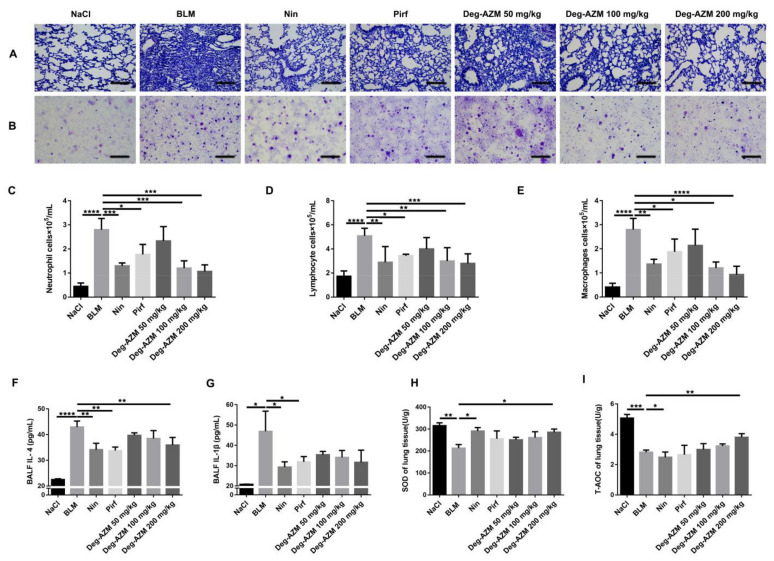
Deglycosylated azithromycin suppresses the inflammatory response and oxidative stress *in vivo*. (**A**) H&E staining of left lung tissues in each group. (**B**) H&E staining of inflammatory cells in the BALF. (**C**–**E**) Neutrophils, lymphocytes and macrophages from BALF in each group were counted on day 7. (**F**,**G**) Effects of deglycosylated azithromycin in Figure 4 and IL-1β in BALF. Cytokines in BALF were determined using ELISA. (**H**,**I**) Effects of deglycosylated azithromycin oxidative stress induced by BLM. The experiments were performed with three technical replicates and two biological repeats. Data are presented as the means ± SD (*n* = 8). Comparisons between groups were achieved using one-way analysis of variance (ANOVA). * *p* < 0.05, ** *p* < 0.01, **** *p* < 0.0001.

**Figure 4 molecules-26-02820-f004:**
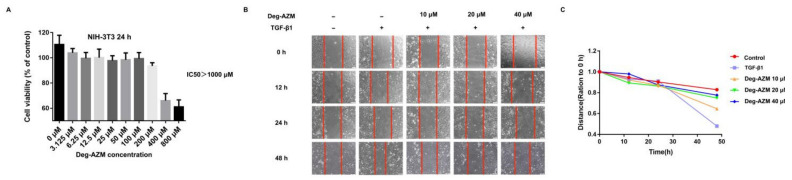
Deglycosylated azithomycin inhibits TGF-β1-induced fibroblast proliferation and migration. (**A**,**B**) MTT assays of NIH3T3 cells. Cells were exposed to the indicated doses of deglycosylated azithromycin for 24 h, IC_50_ = 1078 μM, (*n* = 3 per group). (**C**) Wound healing assays of cells co-cultured with TGF-β1 (5 ng/mL) and deglycosylated azithromycin (10, 20 and 40 μM). The wound closure was photographed at 0, 12, 24 and 48 h post-scratching. Data are presented as the means ± SD (*n* = 3). Comparisons between groups were achieved using one-way analysis of variance (ANOVA).

**Figure 5 molecules-26-02820-f005:**
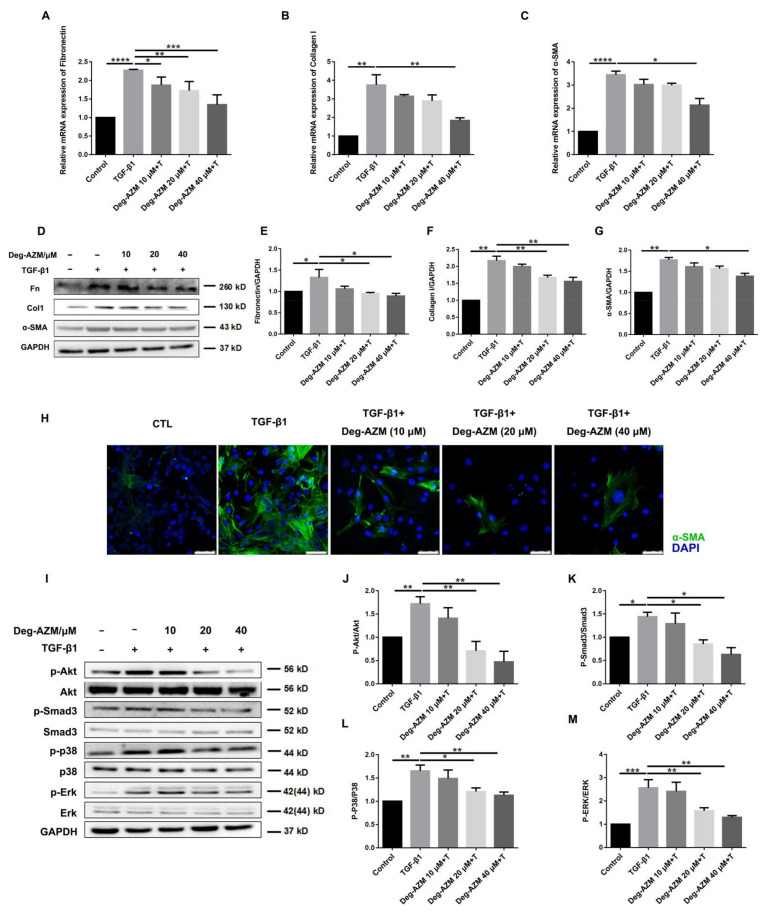
Deglycosylated azithromycin attenuates TGF-β1-induced myofibroblast activation. (**A**–**C**) NIH-3T3 cells were treated with TGF-β1 (5 ng/mL) and deglycosylated azithromycin (10, 20 and 40 μM) for 24 h. Fn, Col1 and α-SMA were analyzed by real-time PCR in NIH-3T3 cells, (*n* = 3 per group). (**D**) NIH-3T3 cells were treated with TGF-β1 (5 ng/mL) and deglycosylated azithromycin (10, 20 and 40 μM) for 24 h. NIH-3T3 cells were extracted for Western blot analysis of Fn, Col1 and α-SMA. (**E**–**G**) Densitometric analysis of Fn, Col1 and α-SMA, using GAPDH as the internal reference. (**H**) Immunofluorescence staining of α-SMA was performed on NIH-3T3 cells treated with/without TGF-β1 (5 ng/mL) and/or deglycosylated azithromycin. (10, 20 and 40 μM) for 24 h. Scale bar = 20 μm. (**I**) NIH-3T3 cells were treated with TGF-β1 (5 ng/mL) for 30 min and extracted for Western blot analysis of p-Akt, p-Smad3, p-p38 and p-Erk1/2 expression. (**J**–**M**) Densitometric analysis of p-Akt, p-Smad3, p-p38 and p-Erk1/2 normalized to Akt, Smad3, p38 and Erk1/2 levels. Data are presented as the means ± SD (*n* = 3). Comparisons between groups were achieved using one-way analysis of variance (ANOVA). * *p* < 0.05, ** *p* < 0.01, *** *p* < 0.001, **** *p* < 0.0001.

**Figure 6 molecules-26-02820-f006:**
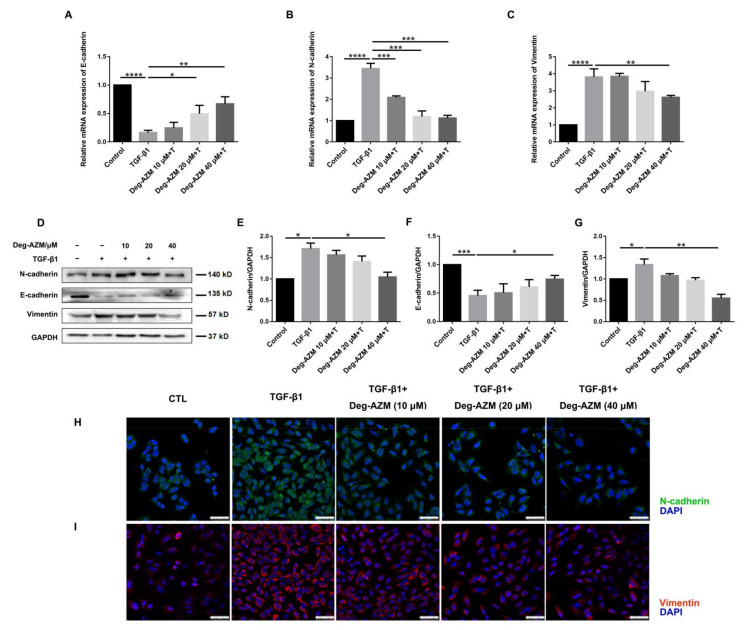
Deglycosylated azithromycin inhibits TGF-β1-induced EMT in epithelial cells. (**A**–**C**) A549 cells were exposed to deglycosylated azithromycin for 30 min (10, 20, and 40 μM) and treated with TGF-β1 (5 ng/mL) for 24 h. The mRNA levels were detected: E-cadherin, N-cadherin and Vimentin were tested by RT-PCR in A549 cells, (*n* = 3 per group). (**D**) A549 cells were treated with TGF-β1 (5 ng/mL) and deglycosylated azithromycin (10, 20 and 40 μM) for 24 h. Protein expression levels of N-cadherin, E-cadherin and Vimentin were assessed by Western blot. (**E**–**G**) Densitometric analysis of N-cadherin, E-cadherin and Vimentin, using GAPDH as the internal reference. (**H**,**I**) A549 cells treated with TGF-β1 (5 ng/mL) and deglycosylated azithromycin (10, 20 and 40 μM) for 24 h were immunostained with N-cadherin and Vimentin. Scale bar = 20 μm. Data are presented as the means ± SD (*n* = 3). Comparisons between groups were achieved using one-way analysis of variance (ANOVA). * *p* < 0.05, ** *p* < 0.01, *** *p* < 0.001, **** *p* < 0.0001.

**Figure 7 molecules-26-02820-f007:**
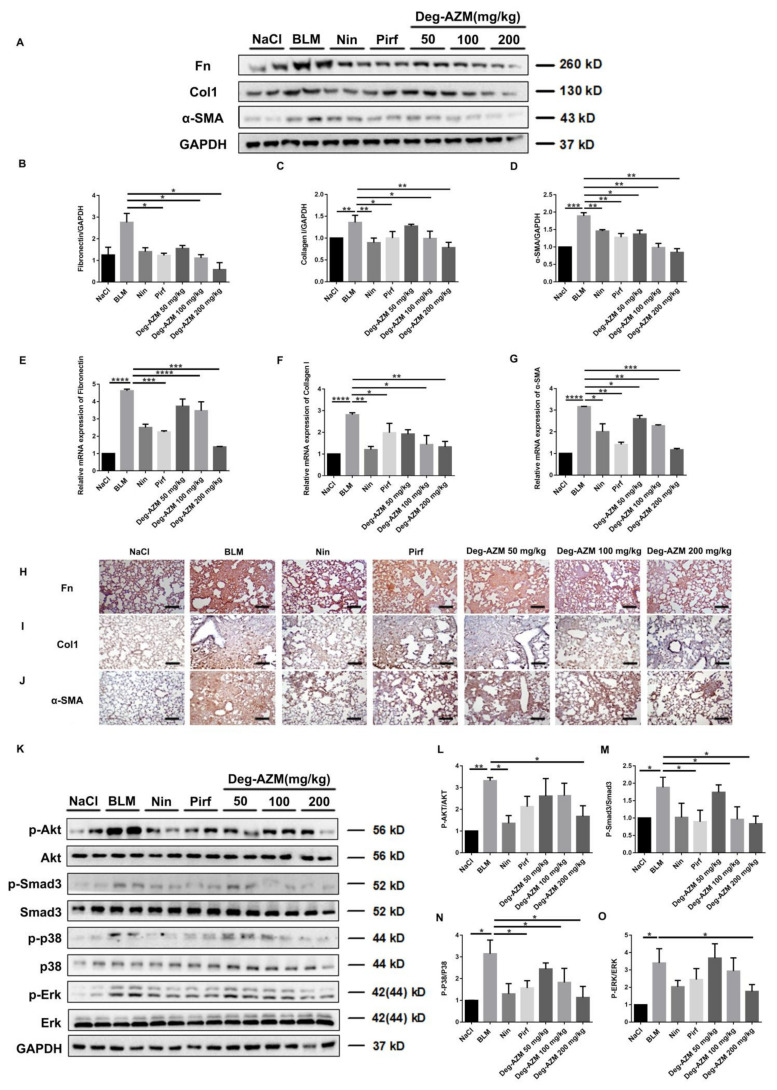
Deglycosylated azithromycin inhibits the fibrogenic activation of pulmonary fibroblasts *in vivo*. (**A**) Western blot analysis of the protein levels of Fn, Col1 and α-SMA in lung tissues. (**B**–**D**) Densitometric analysis of Fn, Col1 and α-SMA, using GAPDH as the internal reference. (**E**–**G**) Real-time PCR was performed to detect the mRNA levels of α-SMA, Col1 and Fn in lung tissues. (**H**–**J**) Immunohistochemical staining of α-SMA, Col1 and Fn in lung tissues. Scale bar = 50 μm. (**K**) Western blot analysis of the protein levels of p-Akt, p-Smad3, p-p38 and p-Erk1/2 expression. (**L**–**O**) Densitometric analysis of p-Akt, p-Smad3, p-p38 and p-Erk1/2 normalized to Akt, Smad3, p38 and Erk1/2 levels, respectively. Data are presented as the means ± SD (*n* = 8). Comparisons between groups were achieved using one-way analysis of variance (ANOVA). * *p* < 0.05, ** *p* < 0.01, *** *p* < 0.001, **** *p* < 0.0001.

**Figure 8 molecules-26-02820-f008:**
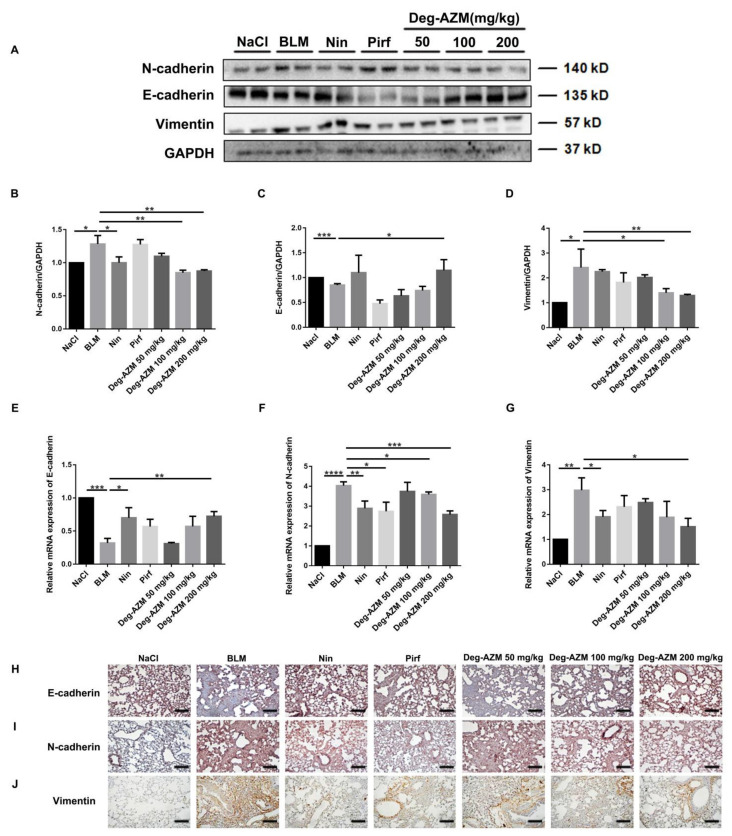
Deglycosylated azithromycin inhibits the EMT phenotype *in vivo*. (**A**) Western blot was used to analyze the protein levels of N-cadherin, E-cadherin and Vimentin in lung tissues. (**B**–**D**) Densitometric analysis of N-cadherin, E-cadherin and Vimentin, using GAPDH as the internal reference. (**E**–**G**) Real-time PCR was performed to detect the mRNA levels of E-cadherin, N-cadherin and Vimentin in lung tissues. (**H**–**J**) Immunohistochemical staining of E-cadherin, N-cadherin and Vimentin-positive cells in the lungs. Scale bar = 50 μm. Data are presented as the means ± SD (*n* = 8). Comparisons between groups were achieved using one-way analysis of variance (ANOVA). * *p* < 0.05, ** *p* < 0.01, *** *p* < 0.001, **** *p* < 0.0001.

**Figure 9 molecules-26-02820-f009:**
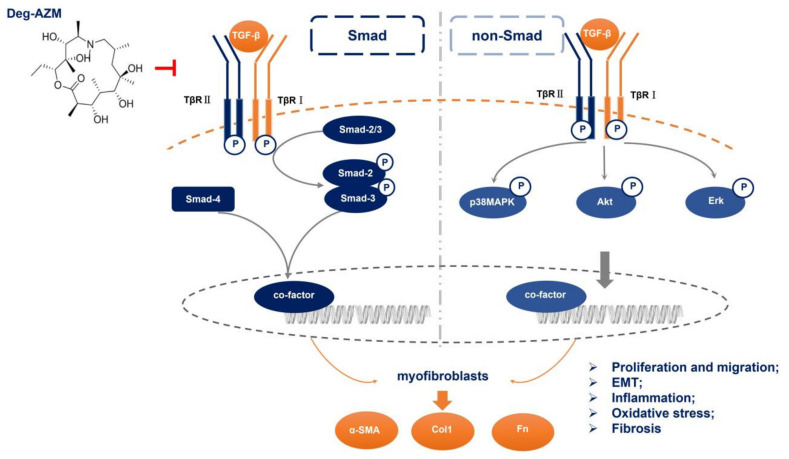
Proposed mechanism for the anti-pulmonary fibrosis effect of Deglycosylated azithromycin. Deglycosylated azithromycin targeting TGF-β1 downstream Smad3 and MAPK signaling pathway, and then inhibiting fibroblasts activation and EMT process, subsequently ameliorates bleomycin-induced pulmonary fibrosis in mice.

**Table 1 molecules-26-02820-t001:** Primers sequences for real-time PCR.

Gene	Forward Primer Sequence (5′–3′)	Reverse Primer Sequence (3′–5′)
Mouse GAPDH	AGGTCGGTGTGAACGGATTTG	GGGGTCGTTGATGGCAACA
Mouse α-SMA	GTCCCAGACATCAGGGAGTAA	GTCCCAGACATCAGGGAGTAA
Mouse Fibronectin	TCGGATACTTCAGCGTCAGGA	TCGGATACTTCAGCGTCAGGA
Mouse Collagen I	ATGTGGACCCCTCCTGATAGT	ATGTGGACCCCTCCTGATAGT
Mouse E-cadherin	CAGCCTTCTTTTCGGAAGACT	GGTAGACAGCTCCCTATGACTG
Mouse N-cadherin	CTCCAACGGGCATCTTCATTAT	CAAGTGAAACCGGGCTATCAG
Mouse Vimentin	GCTGCGAGAGAAATTGCAGGA	CCACTTTCCGTTCAAGGTCAAG
Human GAPDH	GGAGCGAGATCCCTCCAAAAT	GGCTGTTGTCATACTTCTCATGG
Human E-cadherin	ATTTTTCCCTCGACACCCGAT	TCCCAGGCGTAGACCAAGA
Human N-cadherin	TTTGATGGAGGTCTCCTAACACC	ACGTTTAACACGTTGGAAATGTG
Human Vimentin	AGTCCACTGAGTACCGGAGAC	CATTTCACGCATCTGGCGTTC

## Data Availability

The data presented in this study are available on request from the corresponding author.

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
