# Peer review of "Deglycosylated Azithromycin Attenuates Bleomycin-Induced Pulmonary Fibrosis via the TGF-β1 Signaling Pathway"

_molecules, 2021, doi:10.3390/molecules26092820_

Round 1
Reviewer 1 Report
The manuscript “Deglycosylated azithromycin attenuates bleomycin-induced pulmonary fibrosis via the TGF-B-1 signaling pathway” investigates a new anti-fibrotic drug as a candidate for IPF in pre-clinical animal study. Authors have analyzed the efficacy of this drug from histological, molecular, mechanical and immunostaining perspectives. In addition to the in vivo studies, in vitro analysis of TGF-B signaling, fibroblast proliferation and EMT analysis have been made. The quality of the manuscript is further improved by the utilization of pirfenidone and Nintedanib as positive controls.
Few comments are enlisted below.
- The type of ANOVA performed (one-way or two-way) should be enlisted in the legend of each figure.
- Authors identified TGF-B signaling as the critical target for their model. However no data is provided for TGF-B levels in animals exposed to BLM and treated with either saline, Pirfenidone, Nintedanib or Deg-AZM. Authors are invited to analyze TGF-B in either BALF or Tissue for each group.
- Authors have analyzed canonical and non-canonical TGF-B signaling pathways. However Smads, TGFb receptor and either ERK and AKT are client proteins of Heat shock Protein 90, a highly expressed and conserved chaperone. Recent data suggest that HSP90 has a crucial role for fibrosis. The manuscript would benefit from the analysis of HSP90 and its activated (Phosphorylated) form. (https://erj.ersjournals.com/content/49/2/1602152)
- Figure 5 I: pERK western blot contains only one band, while two bands are usually observed for pERK. Please, explain what type of band densitometry has been performed and why.
- More clarity should be addressed for Bleomycin instillation technique. From its description in the methods it is not clear whether the authors inject BLM into the trachea through the skin or perform a blind intubation. Authors should describe in detail the utilized technique and how they have a confirmation that BLM is injected in the trachea and not in the esophagus.
- The quality of the Histology, immunostaining and immunohistochemistry figures is low. Please provide the high-quality images for it.
- The Discussion is brief. The manuscript would benefit from a more throughout argumentation of the results in comparison with other similar studies.
- The graph 2A displays the weight ratio of mice instilled with Bleomycin and treated after the 7th day with either Pirfendone, Nintedanib or diverse dosages of Deg-AZM. Each data point represents the average of 8 animals per group. Standard deviation or standard error mean should be added to the figure. However, important changes in body weight are observed from day 3 to day 6 between groups. At day 6 (1 day before the treatment began), animals of the group BLM lost 16% of body weight (~3.5 gr) while animals similarly instilled with BLM, but that belong to the Nintedanib group only 6% (~1.32). This data is of great concern as a significant difference is clinically observed before the start of treatment. Authors are invited to provide an explanation of why virtually identical animals, that however belong to different groups, display such clinical differences.
Author Response
Dear reviewer,
We sincerely thank the reviewer for the time and efforts reviewing this paper. We have made modifications according to the comments given. Please see the attachment.
Best Regards,
Hao Ruan

Reviewer 2 Report
Ruan et al. demonstrated that deglycosylated azithromycin (Deg-AZM) reduced bleomycin-induced pulmonary fibrosis and inhibited TGF-b1 induced EMT and fibroblasts activation and differentiation.
- While authors claimed that Deg-AZM has no antibacterial activity, AZM should be included in in vivo and in vitro experiments as control for antifibrotic effects.
- The association between TGFb1 and bleomycin-induced pulmonary fibrosis should be addressed and examined.
- The effect of Deg-AZM in inhibiting TGFb1 signaling in vivo (Fig. 7 and Fig. 8) should be examined.
- The authors claimed that TGFb1 signaling as novel target for Deg-AZM. What is the molecular mechanisms? Does Deg-AZM interact with TGFb1 signaling directly or indirectly?
- Zhong et al. demonstrated that Deg-AZM targets transgelin to promote interstinal smooth muscle function (iScience. 2020 Aug 14;23(9):101464.). Does that indicate Deg-AZM contains multiple targets? Additionally, what is the effect of transgelin, a direct target of TGFb signaling, in bleomycin-induced pulmonary fibrosis?
Author Response

(The authors gave the same response as above.)

Reviewer 3 Report
In this study, Hao Ruan et al. evaluated the effect of deglycosylated azithromycin on bleomycin-induced pulmonary fibrosis. The overall manuscript is well organized and written with minimum typographic errors.
I have only minor comments:
- you have big gap/space between line 19 and 20.
- line 66: you at the first time used abbreviation of EMT without prior introduction. Please correct.
- Figure 1C: It will be better unify Y axis with Fig.1D-F and use full name “forced vital capacity“
- Figure 4A: add unit of Deg-AZM concentration (X axis).
- line176: „Densitometric analysis of p-Smad3, p-Erk1/2, p-p38, and p-Akt normalized to Erk1/2, p38, and Akt levels.“ Add Smad3 after normalized to ...
- FIgure 6 I: Correct the red Vinmetin to Vimentin
- Figure 7 I: unify abbreviation of collagen 1 – Col-1 vs: Col1
Author Response

(The authors gave the same response as above.)

Reviewer 4 Report
In this study the authors demonstrated that deglycosylated AZM could reduce lung fibrosis in mice induced by bleomycin. The authors presented adequate data by measuring matrix protein expression and inflammatory related index to show the drug can inhibit the inflammation and fibrosis caused by the bleomycin injury.
The most concerning part is the methods, many wording seems were copied from previous studies, and there are materials in the methods which were not used in this study. For example, CACG-NIH3 cells were not used in this study; no human cells were used, while primer sequences of human were listed.
Although the data for the most part is convincing, the data for signal pathway changes are not strong; last two figures (Fig 7 and 8) used whole lung tissues to show changes in fibroblasts or epithelial cells are weak.
- Please check reference appropriately, some of the statement should cite related reference. For example, p.2, line 53. No reference was cited for the proposed idea.
- Figure 2C - Statistics of lung fibrosis area – please be clear how this was calculated.
- Figure 4A. What is the unit of Deg-AZM concentration unit on x-axis? Fig4B. needs better pictures, the quality is too low.
- Figure 7D. From the WB, FN is hardly decreased, may even increased slightly in the WB shown in A. The densitometry should be double checked. Also, in p.8 line 206, should present FN result separately from a-SMA and Col1.
- Figure 7 and 8 used whole lung tissues to check fibroblasts or epithelial cells markers are not convincing.
- many mistakes in the methods, below are few:
- Liposomal transfection 309reagent were from YEVSEN (Shanghai, China) – where used this?
- (CAGA)12-Lux reporter sta-314bly transfected NIH3T3 cells(CAGA-NIH3T3) – not seen in the paper either
- Not clear how did the cells were treated – p12. Line 322-325
- What age are the mice used this study? As young and aged mice have very different reactions to bleomycin injury.
- FYI, Hydroxyproline should be measured with whole lung, as the fibrosis can be spotty.
- There is no human cells, why list human primers?
Author Response

(The authors gave the same response as above.)

Reviewer 5 Report
This paper evaluates the effects of a deglycosylated azithromycin compound on TGF-beta-induced fibrosis. The authors applied a large number of techniques to test their hypothesis (in vitro and in vivo models). Overall, this is a well-written paper, but the results shown do not seem to fully support the conclusions, as currently described in the manuscript. The results section should simply describe the data shown, rather than over-interpret the data and may be misleading. Revision of the text and figures are needed.
-Results are described out-of-order compared to the figures (line 79). Please check throughout and revise the figure panels or text.
-Add fold-changes and p-values to all results described, rather than qualitative comparisons.
-Please modify the experimental scheme to include and describe which groups were used as the negative and positive controls. Also, be explicit within the text regarding whether an uninjured control was included.
-No dose-dependency was found at the tested concentrations for the functional readouts, only in the histology. This seems consistent with multiple experiments.
-Include statistical analysis of all quantifications within the figures (ex: Fig4B missing stats).
-Please include molecular weight markers for all western blots, as well as show blots in sequence of the protein molecular weight, per convention (high to low mw).
-P-SMAD is not increased with TGF-beta1 treatment based on the representative blot shown in Fig5.
-No significant difference is apparent in the data (only a difference between the highest and lowest concentrations). This claim should be corrected throughout the manuscript or supported with additional data.
Author Response

(The authors gave the same response as above.)

Round 2
Reviewer 1 Report
The authors have followed reviewer's comments and performed additional experiments.
Author Response
Thank you again for your questions and suggestions for this manuscript. Under your guidance, we will strive to do better in future scientific research.
Reviewer 2 Report
The authors have answered most of my concerns in revised manuscript.
Author Response

(The authors gave the same response as above.)

Reviewer 4 Report
Some concerns are not addressed, but at least offered better description of the methods. The authors should do a better job to go through the paper to add appropriate references, not just added only one that the reviewer has pointed out.
Other than that, I am OK with the revision.
Author Response
We sincerely thank the reviewer for the time and efforts in reviewing this manuscript. We have made another revision based on the comments provided, please review.

Reviewer 5 Report
The authors have addressed this reviewer's major concerns. The addition of supplemental material is appreciated.
Minor - It is recommended to remove the references to reviewer comments in the figure legends within the supplemental (Supp. fig 35-40) (Example: Supplementary Figure 35. The entire original gel of TGF-β1 in Response to Reviewer 1 Comments Figure A.)
Author Response

(The authors gave the same response as above.)
